# Predicted ripple dimensions in relation to the precision of in situ measurements in the Southern North Sea

Knut Krämer[1] and Christian Winter[1]

[1]MARUM - Center for Marine Environmental Sciences, University of Bremen, Leobener Str., D-28359 Bremen, Germany

*Correspondence to:* Knut Krämer (kkraemer@marum.de)

**Abstract.** Ripples are common morphological features in sandy marine environments. Their shapes and dimensions are closely related to local sediment properties and the forcing by waves and currents. Numerous predictors for the geometry and hydraulic roughness of ripples exist but due to their empirical nature, they may fail to properly reflect conditions in the field. Here, in situ measurements of tide and wave generated ripples in a shallow shelf sea are reported. Discrete and continuous methods for the extraction of ripple dimensions from digital elevation models (DEM) are inter-compared. The range of measured ripple dimensions is quantified and compared to results of empirical predictors.

The precision of measurement of bedform dimensions is taken as the repeatability of a measurement for inactive conditions and the accuracy of measurement is assessed via comparison to predicted dimensions. Results from field data show that the precision of measurement is limited to 10% of the absolute ripple dimensions. Ripple heights can be estimated with the predictors of Baas (1994) or Soulsby et al. (2012) and lengths can be estimated with predictor of Yalin (1985). The application of different methods for detection of heights may result in derived form roughness heights by up to a factor of two between the traditional statistical estimate and a full evaluation of the spatial bathymetry.

## 1 Introduction

Small scale bedforms like ripples are ubiquitous morphological features in sandy coastal and shelf sea environments. Their formation and dynamics are controlled by waves and currents, while their equilibrium dimensions are commonly described to be related to a characteristic sediment grain size. The existence and evolution of ripples play important roles in the interaction between sea bed and water column (Grant and Madsen, 1979; Bartholdy et al., 2015). Hydraulic roughness and extraction of momentum from the mean flow is enhanced beyond the effect of mere grain roughness by several orders of magnitude due to the presence of ripples (Zanke, 1982; Soulsby, 1997). Furthermore, the presence of ripples influences turnover rates of nutrients and pollutants in the benthic environment when compared to a flat seabed (Nelson et al., 2013; Ahmerkamp et al., 2015).

If the threshold of motion (expressed as the critical bed shear stress or the dimensionless Shields parameter) for a characteristic grain size is exceeded, sediment is transported and bedforms develop. As a fundamental understanding of bedform development is still pending and deterministic prediction is not yet possible, equilibrium ripple predictors based on extensive laboratory and field datasets exist for waves (e.g. Soulsby et al., 2012; Nelson et al., 2013), currents (e.g. Soulsby et al., 2012; Bartholdy et al., 2015) or for combined flows (e.g. Li and Amos, 1998). A classification of the type of bedform and corre-

sponding dominant forcing can be made using the ratio of wave and current shear stress. Wave ripples can be subdivided into orbital ripples scaling with wave orbital diameter (Traykovski et al., 1999), anorbital ripples scaling with grain size (Maier and Hay, 2009) and intermediate forms (Clifton and Dingler, 1984). The dimensions of current ripples are usually related to grain size only (Yalin, 1964, 1985). In contrast to dunes, ripple dimensions are described as independent of the flow depth

(see classification in Venditti, 2013); however, by applying a virtual boundary layer concept, Bartholdy et al. (2015) recently demonstrated that water depth may actually be a controlling factor along with grain size and flow velocity.

Under nonsteady forcing conditions, bedforms continuously adjust in shape and eventually migrate. Equilibrium ripple predictors may not capture this adaptation process resulting in limited prediction of ripple dimensions during unsteady periods. Groundbreaking flume experiments on the development and adaptation of current ripples in very fine sand have been carried

out by Baas (1994), introducing an exponential relaxation scheme for the adaptation of ripple dimensions under changing flow conditions. Time-evolving (nonsteady) ripple predictors have only recently been suggested by Traykovski (2007) and Soulsby et al. (2012). These models also employ an exponential relaxation with a given timescale and rate-of-change coefficients during active conditions to allow for smooth transitions of bedform dimensions and also include decay processes due to wash-out and sheet flow based on additional critical shear stress levels as well as bioturbation.

Understanding of the dynamics of in situ ripple fields may be impeded by relict ripples, which are observed under conditions not related to their formation. These may be inactive bedforms during low flow conditions (around slack water in tidal environments, or after a storm), which may additionally decay through bioturbation, i.e. the activity of benthic and demersal fish or burrowing marine organisms (Amos et al., 1988; Soulsby et al., 2012).

A large number of empirical ripple predictors has been derived from data acquired in flume studies, in which the interaction

between physical and biological processes is not taken into account. Hydrodynamic boundary conditions, local sedimentology and (micro-)biological effects in the field however may be different from flume experiments, e.g. in combined current and waves, in tidal environments dominated by periodically changing flow conditions, or in deep sea environments. This makes field data a necessary prerequisite for the understanding, modeling, and assessment of bed conditions (Schindler et al., 2015; Malarkey et al., 2015).

Methods of ripple measurements in laboratory flumes and in the field make use of optical and acoustical instrumentation. Among others, Li and Amos (1998) used underwater cameras in combination with a scale bar to determine ripple wave lengths. Hay and Wilson (1994) and later Hay and Mudge (2005) used rotary side scan sonar images to describe the evolution of bedform wave lengths during storms. Traykovski (2007) used a sector scanning sonar to measure ripple wave lengths while estimating the height of migrating ripples from the time series of a point measurement of the bed level from an acoustic backscatter

sensor. Bell and Thorne (2007) developed a 3D profiling sonar to measure the small scale bathymetry of rippled sea beds. Before that, the same authors employed a 2D scanning sonar to measure ripple dimensions along transects (Bell and Thorne, 1997). Janssen (2004) collected high resolution bathymetry data using a laser line in rectified camera images taken from a moving sledge. Commonly no assessment of measured accuracy is reported in the studies, despite the range of uncertainty in the technical set-up, analysis, and derivation of seabed properties.

In literature, often spatially averaged values of ripple dimensions are reported while the geometric properties of individual bedforms or their statistical distribution are not reported. However, van der Mark et al. (2008) show that even in laboratory experiments with uniform sediment and stationary flow conditions, bedforms dimensions are far from uniform.

Bedform dimensions and shape can change drastically, when the nature of the dominant forcing changes from strong wave to current dominance or vice versa (Amos and Collins, 1978). This paper is focused on active ripples, i.e. bed conditions in which the shape or dimensions of ripples change over the observation time frame or in which they migrate without changing their general shape or orientation. The time required for the adaptation is a function of sediment transport rate and thus related to the excess shear stress induced by waves or currents and the grain size of the sediment. For current ripples, Baas (1993, 1994) showed in a flume study that the adaptation time is a function of the inverse power of flow velocity and ranges from a few minutes to several days. Additionally, bedform heights were shown to adapt faster than wave lengths. His dataset was used to calibrate the empirical rate-of-change parameters in the time-evolving scheme by Soulsby et al. (2012) with two expressions for height and length. For wave induced bedforms, Nelson and Voulgaris (2014) report that bedform height adapt last after wave length and orientation have almost reached a new stable equilibrium. The adaptation time scale for wave ripples is related to the wave period by Soulsby et al. (2012) and the rate-of-change parameter is related to the wave mobility number.

Recently, Malarkey et al. (2015) additionally highlighted that bedform development can be significantly slowed down by low concentrations (<1%) of biologically cohesive extracellular polymeric substances (EPS) in the sediment matrix.

The overall aims of this study are

1. An assessment of the precision of different methods for the detection and measurement of small scale bedforms from high resolution sonar data in a shelf sea environment

2. The comparison of the measurement precision to the dimensions of small scale bedforms calculated by different wave and current ripple predictors

In the following the bathymetry and sedimentary conditions at the study site on a sandy shelf seabed in the North Sea are described. The setup and devices used to measure the relevant data are shortly introduced. Processing steps for different methods to extract bedform dimensions from raw sonar data are detailed. The measured hydro- and morphodynamic data and ripple characteristics collected over two tidal cycles are analyzed. The ranges and error margins determined by the technical specifications of the sensors and different methods employed to derive parameters from raw sensor data are reported. The range of bedform dimensions as a result of different methodology is shown and evaluated. This range is related and assessed with respect to the dimensions derived from ripple predictors. Implications for the calculation of bedform roughness from ripple dimensions are discussed.

## 2 Methods

### 2.1 Study site

Field data were acquired during cruises on RV *Heincke* to the German Bight at station D (54.09118° N, 7.35881° E) of the NOAH project[1] (North Sea Observation and Assessment of Habitats). An autonomous lander was deployed during cruise HE441 (20–28 March 2015) over a period of around 36 hours. The station was also visited during cruise HE447 in June 2015 but the bedforms were inactive then.

Station NOAH-D is located in the inner German Bight in a water depth of 35 m (Fig. 1). Prior to deployment, a survey of the area surrounding the deployment site by multibeam echosounder revealed a flat and featureless bathymetry on the larger scale (500 m radius). The grain size analysis of grab samples taken prior to deployment of the lander showed bed sediments of fine sand with a median grain size $d_{50} = 105$ µm (Fig. 2). Additional grab samples in the surrounding area exhibit spatially homogeneous sedimentary conditions which is supported by spatially homogeneous backscatter intensity in the multibeam data (not shown).

### 2.2 Lander deployments

Intra-tidal hydro- and morphodynamics are observed by the autonomous seafloor observatory *SedObs* (Fig. 3a). The lander was developed as part of the COSYNA project[2] (Coastal Observing System for Northern and Arctic Seas) (Baschek et al., 2016). It consists of a steel frame with a 2×2 m grating platform providing space for battery power supply and the installation of sensors. The platform rests on four slim height-adjustable inclined legs to which further sensors can be attached close to the sea bed. Weighted foot plates provide stable stand, prohibit subsidence and reduce scouring around the legs. For the application described here, the measurement platform was approximately 2 m above the sea floor to minimize distortions of the near-bed velocity profile. The instrumentation comprises optical and acoustic sensors for the measurement of hydrodynamics, small scale bathymetry and environmental conditions such as water temperature, salinity and turbidity. The lander is deployed from shipboard with the help of a launching frame and is recovered by acoustic release of floating buoys.

Minimization of interference with the system under investigation was a key factor in the design process of the lander as a benthic observatory. In contrast to tripod frames, the four-legged structure allows free flow between the legs. During the launch of the lander the heading is monitored to ensure orientation in alignment with the dominant bottom current direction with the help of a tail-fin on the launching frame (Fig. 3a).

Bathymetry data are checked for the development of scour in vicinity of the legs and disregarded if the bathymetry in the central section is affected. However, with the current velocities common to the deployment sites in the open German Bight, such effects were not observed. Flow velocity and turbulence data are evaluated for possible influence by the lander frame or by other devices and removed if any influence is detected (Amirshahi et al., 2016).

---

[1]See project website for station map and background information: www.noah-project.de

[2]www.hzg.de/institutes_platforms/cosyna/index.php.en

## 2.3 Devices and data

The devices used in this study are summarized in Tab. 1. An acoustic Doppler current profiler (ADCP; Teledyne RDI Workhorse Rio Grande 1200 kHz) was used to record the near-bed velocity profile below the lander. The along-beam resolution of the downward-looking ADCP was 0.1 m and the instrument sampled at a frequency of 1 Hz. Additionally, two acoustic Doppler velocimeters (Nortek Vector ADV) recorded point-wise velocity data at two levels (0.12 m and 0.45 m above seabed) with a sampling frequency of 32 Hz. In combination with the pressure signal recorded, the velocity data was used to calculate wave parameters using the PUV method (e.g. Mudge and Polonichko, 2003).

The small scale bathymetry below the lander was recorded by an 1 MHz 3D acoustic ripple profiler (3D-ARP; Marine Electronics Ltd. (2004); Bell and Thorne (2007)). Its pencil-beam sonar transducer with an effective beam width of 1.8° is mounted on a rotating and tilting mechanism in an oil-filled pressure housing. In a stepwise procedure, the sonar is tilted through a preset arc in 0.9° steps, recording along-beam echo intensities from the water column for every ping (Fig. 4a). After completing one swath, the transducer is rotated by 0.9° about the vertical axis and tilted to the arc starting angle to record the next swath. For our applications, the swath arc was limited to 120° because for grazing angles $\gamma < 30°$ the energy backscattered to the sonar transducer rapidly decreases and the bed echo cannot be detected reliably against background noise. With the sensor installed 1.8 m above the seafloor, a circular area of 6.2 m diameter is covered by the scans. With such settings a full bathymetry scan can be acquired in 11:50 minutes, therefore the scan interval, i.e. the sampling rate of the sonar, was set to 12 minutes. The raw echo intensity data were stored in camera raw format (rw2) with an ASCII header containing sensor settings and meta data and a binary data section listing echo intensity values of successive samples, pings and swaths.

In a morphodynamically active environment, the sampling rate of the bathymetry must be faster than the rate of change of the morphology. Furthermore, as the bathymetry scans are not instantaneous snapshots but rather require a certain time to be recorded, consistency within the scanning period needs to be guaranteed. Although not discussed here in detail, bedform migration with displacement rates of up to 3 cmh$^{-1}$ were observed. At a sampling rate of five scans per hour, this results in a maximum migration distance of 0.6 cm between two successive scans which is lower than the selected resolution of the gridded small scale bathymetries.

## 2.4 Bed detection methods

The raw 3D-ARP data are available as a three-dimensional matrix $\mathbf{S}^{i \times j \times k}$, containing the echo intensities for the number of $i$ samples along the beam, $j$ pings along a swath and $k$ swaths of a full scan. The general form of the echo recorded in individual pings exhibits a high echo level close to the sonar transducer due to ringing. This part of the signal within the near range of the sonar is blanked before further processing. With increasing range from the transducer, the backscattered echo level declines due to signal losses to reverberation and scattering in the water column. Near the bed range, a steep increase to a maximum level can be observed, followed by a more gentle decline towards a constant background noise level. Averaged echo shapes for variable grazing angles are illustrated in Fig. 5.

To reduce noise, the raw echo signals are smoothed by a five-point moving average in along-beam direction. The resulting echo intensity profiles are evaluated for the maximum echo, as the bed usually contributes the strongest reflector. The simplest method of bed detection is therefore to pick the maximum echo in the smoothed ping data. However, as marine life or other instrumentation in the sonar beam may also contribute strong reflectors, the water column echo was only evaluated within a certain depth range around the expected bed level.

Threshold-level methods for bed detection in echo data acquired by similar sonars have been implemented by Smyth and Li (2005) and by Lefebvre (2009). These authors detect the bed level at the depth in which a certain percentage of the maximum ping-wise echo intensity $l_{max}$ is exceeded: $l_p \geq 0.6 \, l_{\mathrm{max}}$ (Smyth and Li, 2005) and $l_p \geq 0.8 \, l_{\mathrm{max}}$ (Lefebvre, 2009). These approaches are extended to account for the widening of the along-beam target shape with increasing grazing angles $\gamma$ (see Fig. 5). Therefore, a threshold level as a function of the grazing angle is introduced:

$$l_p = \left[ 1 - \sqrt{(\cos \gamma)} \right] l_{\mathrm{max}} \tag{1}$$

with values ranging from $l_p = 0.7$ at the outer swath beams to $l_p = 1.0$ at the nadir beam.

Apart from the threshold level, further methods using the first and second along-beam derivatives of the echo intensity, echo gradient and echo curvature, were tested. The maximum echo gradient is usually found midway between background noise level and maximum echo intensity in the rising slope of the signal, given that it is resolved by a sufficient number of samples. The maximum in echo curvature represents the onset of the rising slope of the echo intensity signal.

The last approach for bed detection tested is the cross-correlation with an idealized bed echo model. The bed level in a single water column ping is then found at the along-beam range where the cross-correlation of the recorded ping echo and the echo model is maximized. Bell and Thorne (1997) designed a model of the bed echo (target) represented by a sine wave accounting for the acoustic pulse length and the incident angle between sonar lobe and seabed. The model echo is cross-correlated with the echo profiles and the index of maximum correlation denoting the best fit between echo model data determines the bed range. To account for variable environmental conditions in the echo data at hand, 200 samples ($180°$ in $0.9°$ steps) of echoes for every grazing angle were taken into account for every scan. Averaging the individual pings over all swath angles, a data derived echo model without the need to design an idealized echo shape was obtained.

## 2.5   Coordinate conversion and gridding

The beam coordinates of the detected bed level are computed considering the sound velocity and two-way travel time of the sonar signal yielding an along-beam range. Together with the tilt and rotation angles for the corresponding ping and swath, the bed level is described in spherical coordinates $(r, \theta, \varphi)$ which then are transferred to Cartesian coordinates $(x, y, z)$ (see Fig. 4a).

The along-beam resolution can be estimated from the overall beam range and the number of samples. Typical settings are a beam range of $r_{max} = 4$ m and $n_i = 889$ samples; the resulting vertical resolution for the central vertical beam (nadir) is $\Delta_z = 0.0045$ m. The horizontal resolution is controlled by the area of the sonar footprint as well as tilt and rotation steps. With a beam angle of $\beta = 1.8°$ ($\pm 3$ dB points conical, Marine Electronics Ltd. (2004)) and a sonar height of $h_s = 1.8$ m above the

seabed, the nadir beam ensonifies a circular area of $w_f = 0.056$ m diameter. At the maximum grazing angle $\gamma_{max} = 60°$, the total area ensonified over the echo pulse length has a width $w_{f,\parallel} = 0.226$ m in the swath plane. The along-swath beam spacing is set to $\Delta\beta_\parallel = 0.9°$ steps, resulting in an along-swath spacing of $\Delta s_\parallel = 0.028$ m at nadir ($\gamma = 0°$), $\Delta s_\parallel = 0.057$ m at $\gamma = 45°$ and $\Delta s_\parallel = 0.116$ m at the maximum grazing angle ($\gamma = 60°$). The across-swath beam spacing is controlled by the rotational step of $\Delta\beta_\perp = 0.9°$. With the intersection of the outermost beam at $\gamma = 60°$ with the seafloor at $s_{max} = \tan\gamma_{max} \cdot h_s = 3.118$ m it results in a maximum step of $\Delta s_\perp = s_{max} \cdot \tan\Delta\beta_\perp = 0.049$ m. With $\beta = 2 \cdot \Delta\beta$, the scross-swath footprint width is double the across-swath beam spacing.

As the acoustic pulse is most likely reflected by the highest elevation within the sonar footprint, the depth of troughs may be underestimated (Fig. 4b). Assuming a triangular bedform shape and a maximum slope equal to the angle of repose of sand $\alpha = 32°$, the maximum error in underestimating through depths yields $\varepsilon_{max} = 0.5 \cdot w_f \cdot \tan\alpha = 0.017$ m at nadir and $\varepsilon_{max} = 0.070$ m at the outermost beam in our configuration. As ripple troughs are usually more flat, the error is expected to be less pronounced. With a typical aspect ratio (ripple height over length) $\psi = 0.1$ much lower than the angle of repose, the maximum error reduces to $\varepsilon_{max} = 0.003$ m at nadir and $\varepsilon_{max} = 0.011$ m for the outermost ping.

For comparability among successive scans, the scattered data points were gridded resulting in digital elevation models (DEM) with consistent grid cells (Fig. 3c). With a minimum along-swath sonar step size of 0.028 m at nadir, a grid horizontal grid resolution of $\Delta x = \Delta y = 0.025$ m was selected to maintain the high resolution in the center of the recorded bathymetry even if the effective resolution decreases with increasing beam footprint and spacing towards higher grazing angles.

In the last processing step, the bathymetry is cropped to the central area of 2 m by 2 m for further evaluation of bedform characteristics. This limitation is made because the area outward of the lander legs is shadowed from the sonars field of view and scouring may influence the area in vicinity of the feet. Additionally, the maximum grazing angle for the cropped area is limited to $\gamma = 30°$, reducing the effects of increasing beam spacing and sonar footprint. To better distinguish local ripple features, the global trend of the larger scale surrounding bathymetry is computed from the average bathymetry of all scans of a deployment and removed. The resulting residual zero-mean bathymetry is evaluated by the following methods.

## 2.6 Ripple geometry

Ripple geometry can be described by the orientation $\varphi$ of crests lines in the horizontal plane and the cross-sectional dimensions; height $\eta$, wave length $\lambda$ and aspect ratio $\psi = \eta/\lambda$. The dominant forcing can be distinguished from ripple cross-sectional shape: In contrast to symmetric wave ripples, current ripples exhibit a steeper downstream (lee) slope and a more gently inclined upstream (stoss) slope. A classification for a number of transitional forms between pure wave and current ripples is given by Amos et al. (1988). The ratio of the stoss and lee slope lengths (symmetry index) can be used to identify bedform orientation with regard to the dominant forcing and indicate migration in this direction (Knaapen, 2005).

The geometry of the ripples is extracted from the gridded bathymetry datasets. First, the crest-transverse orientation $\theta$ of the ripple field is derived: The gridded datasets are transferred into binary image matrices using a threshold equal to half the standard deviation of the global elevation $z_{tr} = 0.5\ \sigma_z$ (Fig. 4c). The binary images are processed using 8-connected neighborhoods to identify crest areas of individual bedforms. The detected objects are represented by ellipses of equal area.

Small and circular objects are removed by means of minimum area and ratio of the ellipses semi-axes. The average orientation of the remaining objects is used as characteristic ripple orientation. Figure 6 shows an example for the cropped bathymetry, the binary image with detected ripple orientation and the corresponding distribution of crest-perpendicular orientation in the polar histogram. The precision in orientation detection throughout successive scans, even for inactive bedforms is in the order of $10°$. To avoid abrupt changes in the subsequent computation of bedform height and length, the ensemble average ripple orientation is computed for the complete deployment period, given that it does not change significantly over time. Afterwards, the scans are rotated using the average ripple orientation and re-interpolated to the original Cartesian grid for extraction of ripple dimensions using the following three methods:

1. Statistical method ($\eta_{m,s}$)

   The first is a statistical estimate using the distribution of bed elevations. The standard deviation of elevation, multiplied by a factor $k = 2\sqrt{2}$ was used to estimate bedform heights $\eta_s$ by Traykovski et al. (1999) and Smyth and Hay (2002). This method is usually employed to compute root mean square wave height from water level records and assumes sinusoidal bedform cross sections.

2. Image extrema method ($\eta_{m,i}$)

   The second method finds local extrema in the 3D bathymetry as grid cells surrounded by cells of lower (crest) or higher elevation (trough), similar to finding extreme pixel values in raster images. The averaged ripple $\eta_i$ height is computed from the range between crests and troughs.

3. Transect method ($\eta_{m,t}$, $\lambda_{m,t}$)

   For the third method, transects are defined perpendicular to the crest orientation and evaluated for local extrema (crest and trough) between zero up- and down-crossings. The computed bedform height $\eta_t$ is the average range between the elevations of detected maxima and minima per transect. Apart from height, bedform length $\lambda_t$ is also computed by the transect method as the average along-transect distance between two successive crests. With the DEM spacing and cropping window size used, a total of 80 transects of 2 m length are evaluated.

Methods for evaluation of bedform dimensions can be divided in continuous and discrete approaches. While statistical methods evaluate the continuum of the bathymetry, discrete or direct methods provide dimensions of a limited number of features detected with a given threshold for height and in case of the transect method also length. As described by Friedrich et al. (2007), the disadvantage of discrete methods is the sensitivity of measured dimensions to the thresholds selected. As an alternative, ripple orientation and length can also be determined from spectra obtained from the 2D discrete Fourier transform (DFT) of the gridded bathymetry (Traykovski, 2007; Lefebvre, 2009; Nelson and Voulgaris, 2014) or from 2D autocorrelation. However, these methods require a certain regularity of the bedforms and were not applied here. Especially when primary and secondary bedforms are present, a carefully calibrated direct approach may be more useful than a statistical approach (cf. van der Mark et al., 2008; Ernstsen et al., 2010). The disadvantage of direct approaches is that the selection of thresholds and filter window sizes introduces a certain subjectivity and influences the resulting statistics of bedform dimensions. The

advantage of the direct methods is that they capture a range of bedform dimensions and therefore yields not only average values for the overall bathymetry but also a distribution of dimensions which allows for a statistical evaluation.

## 2.7 Predictors for ripple dimensions

A number of predictors for wave and few for current ripple geometry exists in literature. A recent overview and evaluation of the performance of wave ripple predictors with an extensive dataset from lab and field experiments can be found in Nelson et al. (2013). Soulsby and Whitehouse (2005) present a literature review of predictors for wave, current and combined ripples and Soulsby et al. (2012) recently developed a combined, time-evolving predictor. After determining the dominant forcing, two formulations for wave or current ripples are employed to determine equilibrium height which is then used in an exponential relaxation in the time-stepping procedure (Soulsby et al., 2012).

In contrast to comparison studies as e.g. Nelson et al. (2013) we choose a number of common predictors and compare their range to the range of measured ripple dimensions by the different methods described above.

The following ripple predictors are evaluated with the given median grain size and hydrodynamic data and compared to measured dimensions. The traditional current ripple predictors of Yalin (1964, 1985) (Ya64, Ya85) for length and Flemming (1988) (Fl88) and Baas (1994) (Ba94) for ripple height were selected as they are widely used. For mixed forcing conditions, the recent wave and current ripple predictors of Soulsby et al. (2012) (So12w, So12c) are used by defining the prevailing dominant forcing and selecting the appropriate predictor.

### 2.7.1 Current ripples

Current generated ripple dimensions are usually described as independent of hydrodynamic parameters but scaling with grain size and immersed weight only. An early work by Yalin (1964) (Ya64) predicts current ripple length as

$$\lambda_c = 1000 \cdot d_{50} \tag{2}$$

and was later revised including additional data (Yalin, 1985) (Ya85) in the form

$$600 \cdot d_{50} \leq \lambda_c \leq 2000 \cdot d_{50} \tag{3}$$

While the ratio between bedform height and length may be derived using an empirical relation with the best fit to a large dataset from laboratory and field data by Flemming (1988) (Fl88)

$$\eta_c = 0.0677 \cdot \lambda_c^{0.8098} \tag{4}$$

and the maximum bedform height as

$$\eta_{c,\max} = 0.16 \cdot \lambda_c^{0.84} \tag{5}$$

Baas (1994) (Ba94) gives bedform height as

$$\eta_c = 18.16 \cdot d_{50} \lambda_c^{0.84} \tag{6}$$

Building on this work, Soulsby et al. (2012) (So12c) predict maximum dimensions of current ripples as follows. For height they obtain

$$\eta_{c,\text{max}} = d_{50} \cdot 202 \cdot D_*^{-0.554} \tag{7}$$

and wave length yields

$$\lambda_{c,\text{max}} = d_{50} \cdot (500 + 1881 \cdot D_*^{-1.5}) \tag{8}$$

Equation (7) and Eq. (8) are valid in a range of $1.2 < D_* < 16$, where $D_*$ is the dimensionless grain size

$$D_* = \left[ \frac{g(s-1)}{\nu^2} \right]^{1/3} d_{50} \tag{9}$$

with the density ratio of sediment and water $s = \rho_s/\rho_w$, gravitational acceleration $g$ and kinematic viscosity of water $\nu$. These maximum ripple dimensions are reduced during wash-out conditions and existing ripples are completely eliminated by sheet flow. The different flow regimes are delineated by respective critical Shields parameters. In the measurements presented here, supercritical Shields parameters for bed load transport were found but they remained far below wash-out and sheet flow conditions, thus only the maximum ripple dimensions are used here.

### 2.7.2 Wave ripples

Predicted wave ripple dimensions commonly scale with a dimensionless number derived from wave parameters in relation to sediment grain size and immersed weight. Soulsby et al. (2012) (So12w) found that the use of the ratio of wave orbital amplitude and median grain size $\Delta = A/d_{50}$ as independent variable gives the best representation of a large dataset of measured ripple dimensions from flume and field studies. They use the following empirical predictors for wave induced ripple wave length

$$\lambda_w = \left[ 1 + 1.87 \times 10^{-3} \Delta (1 - \exp(-(2.0 \times 10^{-4} \Delta)^{1.5})) \right]^{-1} A \tag{10}$$

and height

$$\eta_w = 0.15 \, (1 - \exp(-(5000/\Delta)^{3.5})) \, \lambda_w \tag{11}$$

Earlier works as cited in Li and Amos (1998) based on flume and field data predict wave ripple dimensions as follows. Grant and Madsen (1982) (GM82) predict height as

$$\eta_w = 0.22 A (\theta_w/\theta_{cr})^{-0.16} \tag{12}$$

and length as

$$\lambda_w = 6.25 \eta (\theta_w/\theta_{cr})^{0.04} \tag{13}$$

Li et al. (1996) (Li96) give

$$\eta_w = 0.101 A (\theta_w / \theta_{cr})^{-0.16} \tag{14}$$

for height and

$$\lambda_w = 3.6 \eta (\theta_w / \theta_{cr})^{0.04} \tag{15}$$

for length. In Eq. 12 – 15, $\theta_w$ is the wave-induced and $\theta_{cr}$ the critical Shields parameter.

## 2.8 Hydraulic roughness

When bedform dimensions are known, their effect on the flow can be assessed if the hydraulic roughness length is obtained using empirical relations (Li and Amos, 1998; Lefebvre et al., 2016). The impact of form roughness due to bedforms is important for numerical models as it can exceed the effect of grain roughness $k_g$ by orders of magnitude (e.g. Soulsby, 1997). A widely used (bed-)form roughness predictor is defined by van Rijn (1984)

$$k_{s,f} = 1.1 \cdot \eta \cdot (1 - e^{-25\eta/\lambda}) \tag{16}$$

Another common form of roughness length derived from ripple dimensions is $k_f = f(\eta^2/\lambda)$ with height in a power of two over length (see overview in Lefebvre et al., 2011) with varying scaling scaling factors. Soulsby (1997) presents it as follows

$$z_{0,f} = a_r \frac{\eta^2}{\lambda} \text{ (with scaling factor typically } a_r = 1). \tag{17}$$

## 3 Results

### 3.1 Hydrodynamics and sediment mobility

Hydrodynamic and meteorological data from the measurement site for a period of 36 hours are displayed in Fig. 7. Over the tidal cycle, water depths range from 34 m at low tide to 37 m at hight tide. Current velocities measured by the lower ADV 0.12 m above the seabed range from 0.1 to 0.3 ms$^{-1}$. The depth-averaged flow velocities measured by the downward-looking ADCP are 25% higher. The wind direction changes from westerly winds with speeds of up to 15 ms$^{-1}$ during the first day of the deployment through North to Easterly directions with speeds of 5–15 ms$^{-1}$ on the second day. Wave parameters were calculated using the velocity and pressure data from the lower ADV. Significant wave heights range from below 0.5 m in the first half of the measurement up to 2.5 m in the second half with a peak period between 8 s and 10 s.

To relate the hydrodynamic forcing to sediment mobility, Shields parameters were computed for wave ($\theta_w$) and current ($\theta_c$) forcing and the critical Shields parameter ($\theta_{cr}$) was defined for the given median grain size (Fig. 8a). For the first 18 hours of the deployment, conditions with excess shear stress were observed only during peak flood and ebb current. Wave-induced excess shear stress conditions are reached for a period of 4 hours starting around 15:00 h local time on the second day, followed by a period with current-induced excess shear stress lasting for around 4 hours during flood current around 22:00 h local time.

## 3.2 Bed detection

An inter-comparison of the different methods for bed detection shows that all threshold level methods reproduce similar characteristics of the rippled seabed (Fig. 9). They mainly differ in the absolute level of average depth. The maximum echo gradient, maximum echo curvature and 60% maximum echo method (Smyth and Li, 2005) provide a median depth around 0.025 m higher than the median depth computed by the remaining methods. Additionally, the 60% max. method exhibits a slight dependence on the grazing angle and returns a bowl-shaped bathymetry (see Fig. 9b). The comparison of the different bed detection methods revealed that picking the maximum amplitude of a smoothed echo within a certain range of the expected bed level provides the most efficient approach. Level threshold methods do not enhance the bathymetry DEMs and echo gradient and curvature methods are less robust. Bed picking by cross-correlation with an echo model is more computationally expensive than the level threshold methods but it accounts for the shape of the complete bed echo rather than depending on a single value. However, the bed echo model approach is limited to flat seabeds and a perfectly horizontal sonar with a nadir beam normal to the bed, where only the grazing angle determines the incident angle between sonar lobe and bed. For rippled seabeds however, the exact morphology within the sonar footprint needs to be known a priori to adapt the echo shape to the true incident angle. Echo model methods may therefore rather serve as enhancement of the bathymetry computed by a threshold level method in a first run.

## 3.3 Ripple dimensions

Ripples with a wave length of $\lambda_{m,t} = 0.215$ m and a height of $\eta_{m,t} = 0.013$ m (aspect ratio $\psi = 0.06$) are revealed in the bathymetry (Fig. 8b,c). The largest measured bedform heights of 0.019 m are obtained by the statistical method followed by 0.017 m by the image extrema methods whereas the evaluation of extrema in individual grid transects yields the lowest absolute heights of 0.013 m (Fig. 10a). The dimensions remain stable for the first 24 hours of the deployment and the bedforms are considered inactive during this period as $\theta_c < \theta_{cr}$. Thus, the scatter of their measured dimensions is used to quantify the precision of the methods used for their detection. With the increasing flood current velocities and wave action on the seafloor from 24 hours onwards, the ripple height decreases by 0.004 m over a period of 2 hours and increases to the initial height over the following 6 hours with increasing tidal current velocity. No significant changes in ripple wave length can be observed. In terms of height evolution the trend of change of ripple height on the second day of the deployment is captured by all three methods. The statistical method returns the most robust results resulting in less scatter between successive measurements.

In comparison with ripple height predictors, all methods of bedform detection produce values that fall within the range between mean (0.011 m) and maximum (0.024 m) bedform height as given by Flemming (1988) (Fl88). Following Baas (1994) (Eq. 6), the predicted current ripple height equals 0.015 m. Predicted current ripple dimensions using Eq. (7) and (8) from Soulsby et al. (2012) result in $\eta_{p,c} = 0.015$ m and $\lambda_{p,c} = 0.124$ m. Predicted wave ripple dimensions from Eq. (10) and (11) follow the evolution of wave orbital velocities, however waves are expected to be dominant only where $\theta_w > \theta_c$ (Fig. 7a: Day 2, 15:00 h – 19:00 h). During this period, the maximum predicted wave ripple dimensions are $\eta_{p,w} = 0.017$ m

and $\lambda_{p,w} = 0.115$ m. Wave ripple height predicted by Li et al. (1996) results in 0.015 m while the height of wave (orbital) ripples by Grant and Madsen (1982) results in 0.032 m.

Bedform wave lengths derived based on the transect method amount to $\lambda_{m,t} = 0.215$ m (Fig. 10b). The mean length for current ripples predicted (Yalin, 1964) results in 0.105 m and the range predicted by the relations of Yalin (1985) yields 0.063 m – 0.210 m. Wave ripple length predicted by Grant and Madsen (1982) yields 0.205 m while the length predicted by Li et al. (1996) results in 0.054 m and lengths predicted by Soulsby et al. (2012) amount to $\lambda_{p,c} = 0.124$ m for currents and $\lambda_{p,w} = 0.109$ m for waves.

The discrete transect method allows a statistical evaluation of the distribution of measured dimensions. The evolution of dimension histograms along with statistical parameters for bedform height and length are displayed in Fig. 11. Due to the relatively large gridding cell size of 0.025 m the distribution of lengths is rather narrow and mostly varying between two cells (Fig. 11b). The standard deviation of ripple height increases from 0.005 m to 0.007 m throughout the first day and decreases again to 0.004 m with the wave event (18:00 h – 21:00 h on the second day). This may indicate that the bedform become more regular due to the pronounced dominance of waves in this period.

### 3.4 Hydraulic roughness

Roughness lengths $z_{0,f}$ and Nikuradse's equivalent sand roughness $k_{s,f}$ resulting from the different ripple heights and the length from the transect method are summarized in Tab. 2 along with reduction factors with regard to the statistical method. Nikuradse's roughness $k_{s,f}$ (Eq. 16) returned using ripple dimensions from the image extrema method is reduced by a factor of 0.8 and by a factor of 0.6 for the transect method. Due to the squared ripple height in Eq. 17, the difference between the methods is even more pronounced for roughness height $z_{0,f}$ which returns reduction factors of 0.71 and 0.47 for image extrema and transect methods, respectively.

## 4 Discussion and assessment

### 4.1 Methods for dimension measurement

Only one method is shown for the analysis of ripple orientation and length from sonar data, but three methods can be compared for the calculation of ripple heights. The statistical method (e.g. Traykovski et al., 1999) assumes a two-dimensional sinusoidal ripple field and computes its root mean square height. The second method picking regional extrema in the bathymetry only measures the height of a limited number of features. The evaluation of transects makes use of the complete scan at the grid resolution and averages over a larger number of regional extrema along the transects.

If bedform dimensions are computed from transects perpendicular to bedform crests, the result depends on the lateral position of the transect. As can be seen in Fig. 3b, ripples found in the field often exhibit curved crest lines of limited length rather than being purely two-dimensional features. Furthermore, the instantaneously observed rippled seabed always holds a history of varying dominant forcing drivers, magnitudes and directions. Transitional states may comprise newly formed active ripples

superimposed on decaying relict ripples with different orientation. Within a three-dimensional field, any selected transect will cut across individual ripples at an arbitrary position with respect to its lateral elevation profile. The ripple height can only be regarded as meaningful by statistically evaluating multiple transects. This is underlined by van der Mark et al. (2008), who state that bedforms are far from regular features that can be easily described using mean values, even in laboratory flume experiments with uniform sediment and stationary flow conditions.

## 4.2  Precision of measurement

To assess the *accuracy* of the measurement, a priori known topography under controlled laboratory conditions would be required. This cannot be achieved under field conditions. However, the *precision* of the different methods described here, i.e. the repeatability of a dimension measurement, can be estimated from the inter-comparison of the different methods and the temporal variability of the dimensions obtained from each individual method during stationary, inactive periods.

The different methods for ripple height measurement yield different absolute values but the magnitude of the change in height is captured by all three methods. For a better assessment of the precision of the methods, bedform dimensions from the first 18 hours of the deployment were summarized in box plots exhibiting the distribution of ripple height and length during stationary conditions. The results shown in Fig. 10 indicate that both ripple height and wave length can be measured with a precision smaller than 10% of their absolute dimensions, regardless of the method used. The distributions of ripple height for all three methods are negative-skewed. Judging from 25[th] and 75[th] percentiles, the statistical method provides the most narrow range of ripple height while image and transect extrema yield comparable ranges.

As for ripple length, both 2D cross-correlation and DFT did not prove robust thus the transect method remains. Its results fall into the wide range of lengths predicted by Yalin (1985) but is around 60% larger than lengths predicted by Soulsby et al. (2012) for wave ripples and still about 40% larger than length predicted for current ripples scaling with grain size only.

## 4.3  Form roughness

The overestimation of ripple height has a significant effect on the calculation of hydraulic roughness due to the fact that height is used in a power of two in common roughness predictors (see list in Lefebvre et al. (2011, 2016)). While the range of predicted heights is in good general agreement with measured average values, the So12 predictor tends to represent maximum heights of individual ripples rather than an along-crest average height given a certain three-dimensionality with varying crest elevation. If ripple height measured as average over individual transects is compared to the results from the statistical method, it is found that the latter gives values 40% larger than the transect method. This corresponds to a roughness height increase by a factor of 1.56 if the ripple dimensions are used to predict form roughness using bedform roughness height as given by Eq. (16) (van Rijn, 1984) and an increase of roughness height by a factor of 1.96 using the relation given by Eq. (17) (Soulsby, 1997).

## 5    Conclusions

A setup of instruments and data processing methods for field measurements of ripple dimensions and dynamics was described. While the *accuracy* of the measured ripple dimensions cannot be determined without an absolute reference value, both ripple heights and wave lengths can be measured with a *precision* smaller than 10% of their absolute dimensions during inactive conditions. All methods tested are consistent with regard to the ripple dimensions computed. Observed relative changes in height are in the order of several millimeters between successive scans during active periods. When applying one method the dynamics of ripple dimensions, i.e. the relative changes can be reliably obtained and linked to changes in the forcing hydrodynamics.

The overall range of current ripple height can be predicted using the empirical relation by Flemming (1988). The current ripple predictors from Baas (1994) and Soulsby et al. (2012) and the wave ripple predictors from Li et al. (1996) and Soulsby et al. (2012) fit measured heights more closely. Measured ripple lengths compare best to the upper end of the wide range given for current ripples by Yalin (1985) and wave ripples by Grant and Madsen (1982) but are somewhat longer than lengths predicted for both wave- (Soulsby et al., 2012) and current dominated ripples (Yalin, 1964; Soulsby et al., 2012). The measured lengths of the ripples are best predicted by the upper end of the range for current ripples given by Yalin (1985) and wave ripples by Grant and Madsen (1982).

The performance of time-evolving predictors introduced by Traykovski (2007) and Soulsby et al. (2012) could not be evaluated. The predictor of Traykovski (2007) was developed for wave-orbital ripples in more energetic environments. Both predictors could not predict the small range of dynamic evolution of ripple heights in the shelf sea area. This may also be related to the migration of the ripples due to nonlinear interaction of wave and current forcing, which is not covered by the predictors. Additionally, the migration of bedforms observed may result in the opposed trends in the development of ripple heights during the wave dominated conditions (around 18:00 h on the second day) (8a).

The commonly used statistical estimation of ripple height yields heights 40% larger than average heights obtained by the transect method. This results in calculated form roughness height to increase by a factor of two. To account for the spatial variability of ripple heights, dimensions derived from transects should be considered whenever spatial bathymetry data with sufficient resolution are available.

*Author contributions.* C. Winter designed the field campaigns and measurement setup on the *SedObs* lander. K. Krämer and C. Winter collected the data during cruises HE441 and HE447 on board RV *Heincke* with support of the Coastal Dynamics group at MARUM. K. Krämer performed the data processing and analysis the data. K. Krämer and C. Winter wrote the manuscript.

*Acknowledgements.* The lander setup has been supported through the project Coastal Observing System for Northern and Arctic Seas (COSYNA). Data presented here was collected by the MARUM Coastal Dynamics group. Their willingness to suffer in heavy seas is gratefully acknowledged. The authors also appreciate the support from captain and crew on RV *Heincke*. Processed hydrodynamic ADV

data were kindly provided by S.M. Amirshahi. K. Krämer appreciates the support of the BMBF funded research project NOAH – North Sea Observation and Assessment of Habitats and of GLOMAR – Bremen International Graduate School for Marine Sciences.

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

**Table 1.** *SedObs* lander sensors, measured parameters and sampling rates.

| Sensor | Parameter | Sampling rate |
|---|---|---|
| ADCP 1200 kHz (downward) | Flow velocity (profile) $\boldsymbol{u}(z)$ | 1 Hz |
| 2×ADV | Flow velocity (point), Wave parameters | 32 Hz |
| 3D-ARP | Bathymetry $z(x,y)$ | 1 in 12 min. |
| CTD | Conductivity, temperature, pressure $(C, T, P)$ | 1 Hz |
| Digital camera | Underwater photos (first 90 min. of deployment only) | 0.1 Hz |

**Table 2.** Roughness lengths for measured ripple dimensions using Eq. (16) and (17).

| Method | statistical | image extrema | transect |
|---|---|---|---|
| $k_{s,f}$ [-] Eq. (16) | 0.01861 | 0.01486 | 0.01115 |
| reduction with regard to stat. method | 1.00 | 0.80 | 0.60 |
| $z_{0,f}$ [m] Eq. (17) | 0.00168 | 0.00119 | 0.00079 |
| reduction with regard to stat. method | 1.00 | 0.71 | 0.47 |

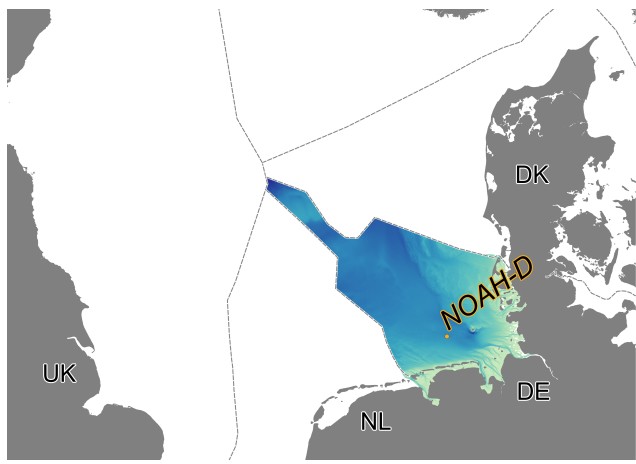

**Figure 1.** Overview map with the location of station NOAH-D in the German Bight.

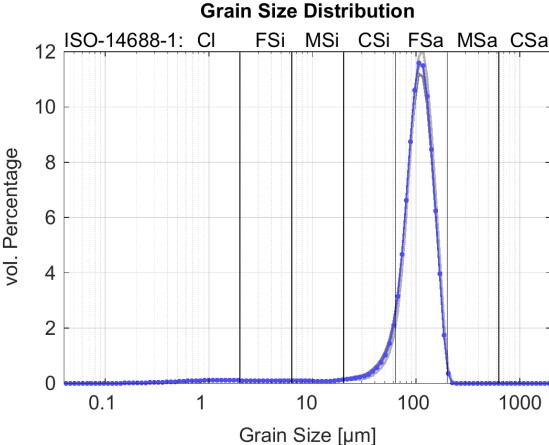

**Figure 2.** Grain size distribution and classification from Coulter laser diffractometer analysis of Shipek grab sample taken at deployment site. Gray curves in the background from grab samples in the surrounding area are shown to indicate spatially homogeneous sedimentology.

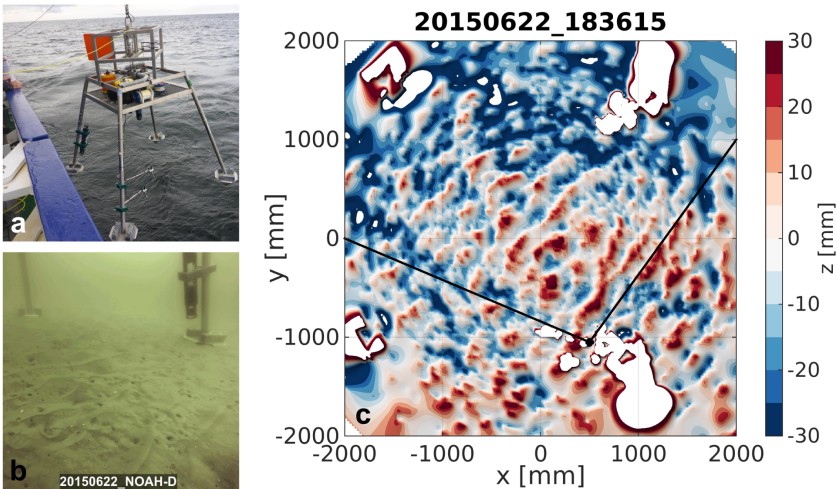

**Figure 3.** (a) Deployment of autonomous sea floor observatory *SedObs*. (b) Underwater photo showing rippled seabed. (c) Cropped sonar image with ripples and lander foot plates visible in the small scale bathymetry. Plane coordinates $(x, y)$ are centered on sonar transducer and elevation $z$ is given as zero-mean. The position and field of view of the camera in (b) are indicated by a black dot and black lines.

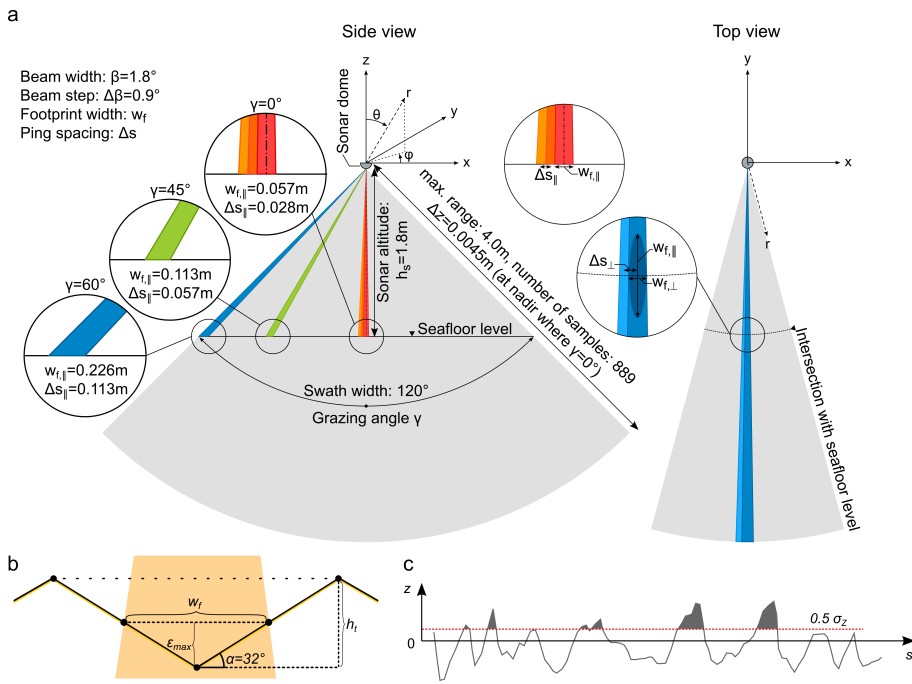

**Figure 4.** (a) Definitions of sonar coordinates and beam footprints on an average seafloor level at different grazing angles. (b) Estimation of maximum error in ripple through depth due to picking of highest elevation within the sonar footprint. (c) Generation of binary image from bathymetry using half the standard deviation of the elevation as cutoff threshold.

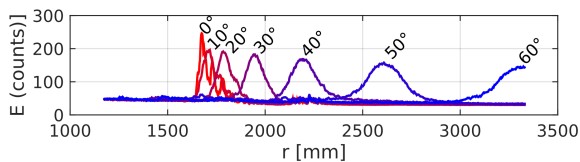

**Figure 5.** Echo intensities over range derived from scan averaged water column echoes for different grazing angles.

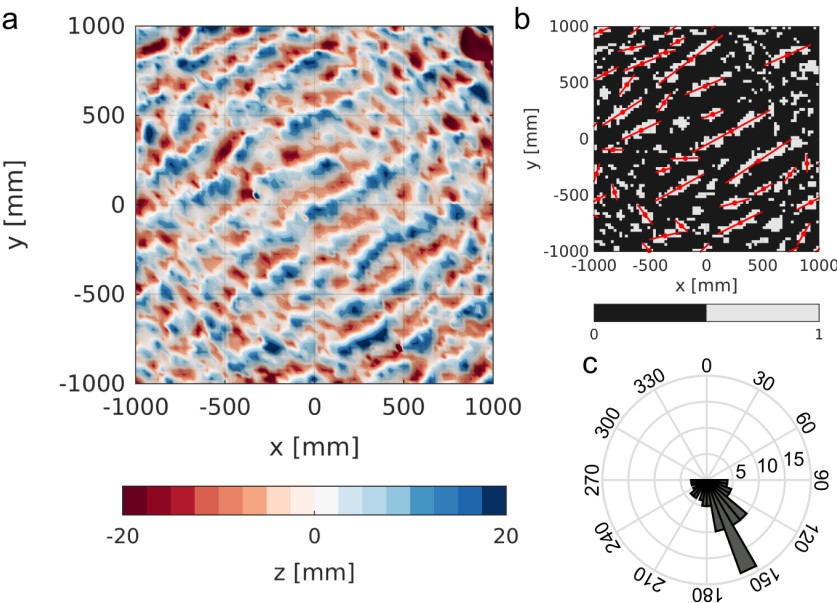

**Figure 6.** (a) Small scale bathymetry cropped to the central 2×2 m below the sonar. (b) Overlay of detected objects in 8-connected neighborhood on binary image with a threshold of $0.5\sigma_z$. Object centers and major axes are marked in red. (c) Polar histogram of ripple crest-perpendicular orientation in degrees from North with percentage of total number of objects on the radial axis.

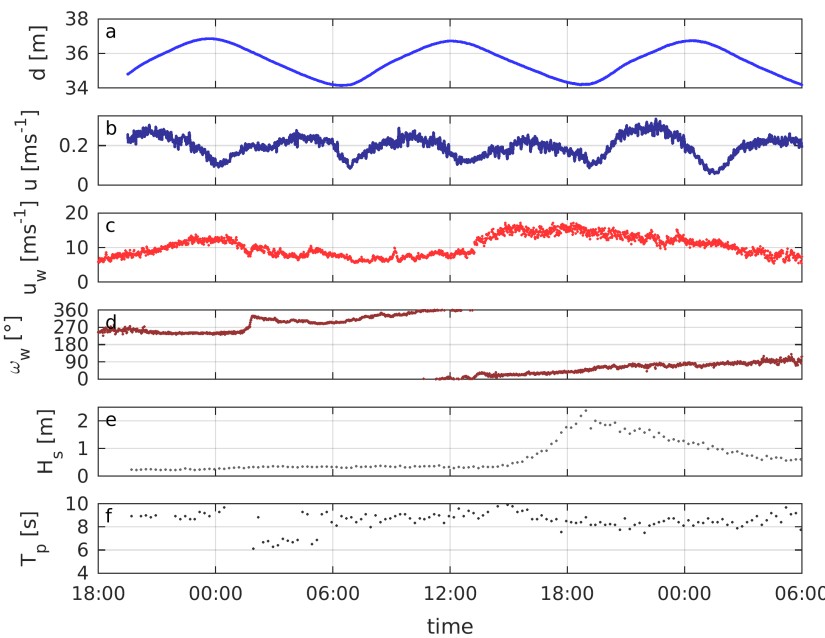

**Figure 7.** Hydrodynamic conditions at station NOAH-D, 20–22 March 2015. (a) Water level, (b) flow velocity at a height of 0.12 m above seafloor (c) wind speed and direction and (d) significant wave height and peak period.

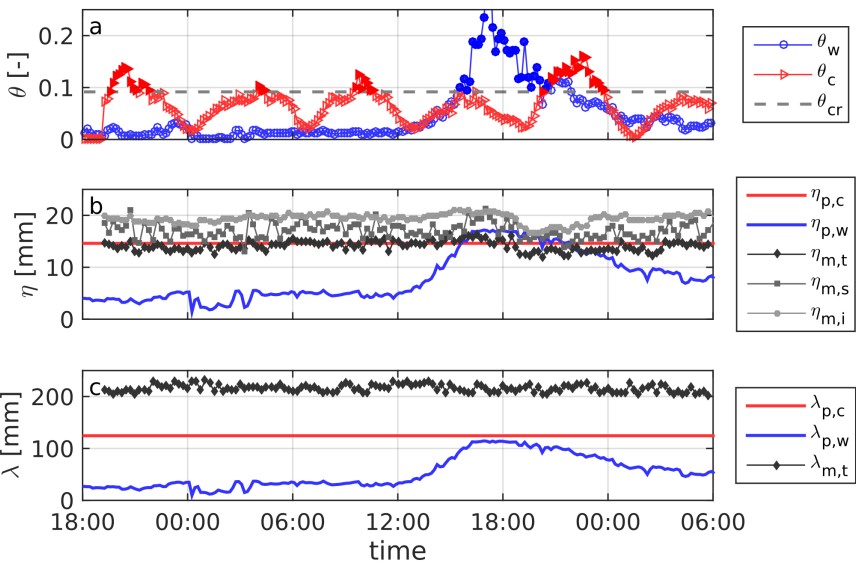

**Figure 8.** (a) Shields parameters for wave orbital velocities $\theta_w$, tidal current $\theta_c$ and critical Shields parameter $\theta_{cr}$. During supercritical conditions ($\theta > \theta_{cr}$), filled markers indicate the dominant forcing. (b) Evolution of ripple height and (c) wave length compared to predicted equilibrium dimension for wave and current forcing as given by Soulsby et al. (2012). Indices in the legends of (b) and (c) indicate: p-predicted, m-measured, c-current, w-wave, t-transect, s-statistical and i-image extrema.

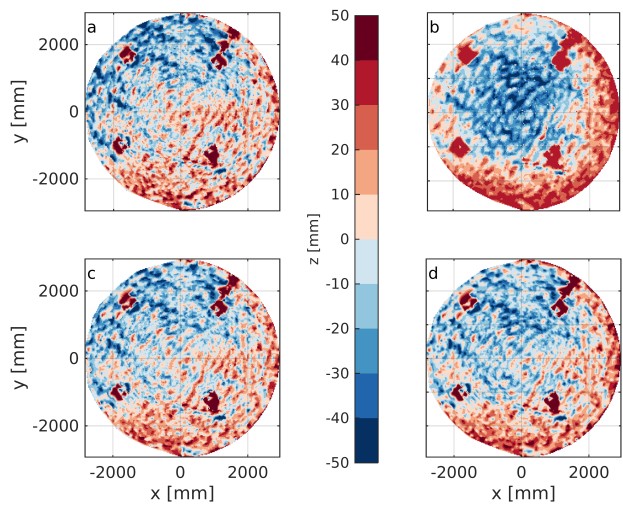

**Figure 9.** Comparison of bathymetries obtained from different bottom-picking methods. (a) Maximum echo, (b) 60% max. echo (Smyth and Li, 2005), (c) 80% max. echo (Lefebvre, 2009) and (d) grazing angle related coefficient of max. echo. Elevation $z$ is given as zero-mean.

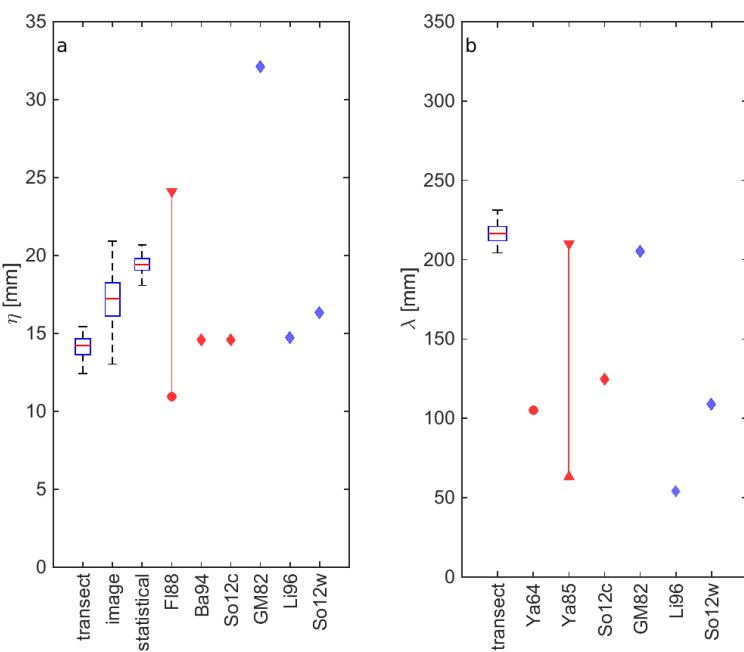

**Figure 10.** Box plots of the precision of measured dimensions during stationary conditions and accuracy in comparison with predicted equilibrium dimension for wave and current dominated conditions using the equations (2–15). (a) Bedform height, measured by 2D transect wise extrema, 3D image extrema and statistical method. Markers indicate predicted ripple heights using the expressions for currents from Flemming (1988), Baas (1994), Soulsby et al. (2012) and for waves from Grant and Madsen (1982), Li et al. (1996) and Soulsby et al. (2012). (b) Bedform wave length measured from 2D transects. Markers indicate predicted ripple length using the current expressions from Yalin (1964, 1985) and Soulsby et al. (2012) and the wave expressions from Grant and Madsen (1982), Li et al. (1996) and Soulsby et al. (2012). In box plots, red line denotes median, blue box indicates 25[th] and 75[th] percentiles and dashed lines extend to extreme values.

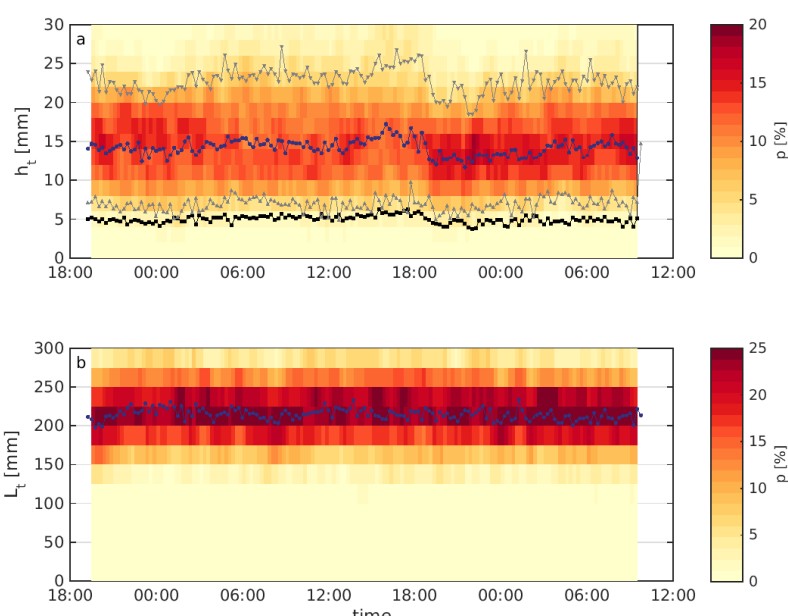

**Figure 11.** Evolution of histograms and statistics of bedform dimensions measured by the transect method. (a) Bedform height and (b) bedform wave length. Blue dot markers indicate median, black squares indicates the standard deviation and gray triangles indicate $5^{th}$ and $95^{th}$ quantiles. Due to the large cell size and narrow distribution of bedform lengths, the latter are only displayed for heights.