# Peer review of "Predicted ripple dimensions in relation to the precision of in situ measurements in the Southern North Sea"

_Ocean Science, 2016_

## Referee Comment (RC1) · Anonymous Referee #1 · 27 May 2016

This submission discusses the estimation of ripple height, orientation and wavelength from detailed measurements of bed topography. The methods of measurement and estimation are as often carried out; the aim here is to assess precision (repeatability) and possible accuracy of the results, especially if conditions are changing so that the statistics of bedforms are not in equilibrium. Comparison is made with predictions from empirical formulae from the literature, for various conditions of waves and currents which were also measured.

This represents thorough work in which the limitations of the measurements and estimates are well discussed, along with the accuracy that "might" be expected. [No absolute estimate of accuracy was possible because there is no known "truth" for the

variables estimated.] It looks to me as though this work should be useful – at least cautionary. However, its value would be clearer to a non-specialist (such as myself) if there were clearer statements of aims (in the Introduction) and how this advances on previous work (in the Discussion).

Although the use of English is generally good (and certainly appears so), there are several sentences where it was a struggle to follow the grammar and where the meaning would certainly be clearer with some simplification and re-ordering. This will have to be done by the authors and not left to copy-editors to guess what is intended.

Detailed comments.

Line 9. I don't like "precision of detection". Either precision of some measurement or threshold of detection.

Lines 11-12. This sentence lacks mention of variations to which "up to a factor of two" applies.

Lines 20-21. "expressed . . parameter" gets in the way of the rest of this sentence. It could be a separate sentence (in parentheses).

Line 16 ". . modelling and assessment . ."

Somewhere near the end of the Introduction (top of page 3) should be a more explicit statement about the aim of this paper (before lines 4-9 saying what was done).

Line15. Unclear why Fig. 1 is referred to here.

Lines 3-5. This sentence is too complex and (I think) grammatically incorrect: "is ex-
ceeded" is redundant given the symbols "$\geq$"? "respectively" is too far from the things being ordered. Break up the sentence, perhaps by defining lp and lmax in a separate sentence.

Line 6. Re-arrange (maybe split) this sentence. I think "where . . time" refers to the target shape and not to nadir.

Lines 12, 13. Echos do not have "slopes". Are you referring to the intensity as a function of distance as plotted in figure 3.

Line 20. "can be" – "was" if this is what you actually did. It reads like a good idea deserving careful description.

Line 4. I don't think $\eta$, $\lambda$ have been defined; they could probably be replaced by words, or bring forward the definitions from line 16.

Line 15. The "phi" symbol should immediately follow "orientation".

Line 30. "every deployment" is unclear. Probably not "deployment" but a briefest statement of what is the "ensemble".

Line 8. "(trough)" (typo).

Lines 10-12. This could be clearer. Is a crest the extreme maximum between any up-crossing and the next down-crossing of zero? If crests are defined dependent on zero-crossings, why not more directly use the average of distance between successive up-crossings of zero (or down-crossings of zero)? Presumably the result would be almost the same.

Lines 28-29. So12w and So12c terminology implies separate predictors for ripples under waves and for ripples under currents. Then (line 29) "are used" but what exactly

is applicable to mixed forcing conditions?

Page 8. Units need to be stated for the dimensionally inconsistent equations (4), (5).

Page 11.

Line 4. ". . more pronounced than for Nikuradse roughness using (11); (12) results in . ."

Line 23. This sentence is incorrect. Replace one "of" by ","?

Figure 8. There are two lines for Fl88 in (a) and Ya85 in (b). Please explain in caption – refer to (4), (5) and (3)?

---

## Referee Comment (RC2) · Anonymous Referee #2 · 29 Jul 2016

Short summary of the MS

Krämer and Winter (KW) have carried out lander-based in situ measurements of ripples in a shelf environment. The two main objectives, as also highlighted in the title of the MS, are 1) to evaluate the precision of the in situ measurements, and 2) to compare the in situ measurements with ripple predictors. KW find that 1) ripple dimensions can be measured with a precision smaller than 10% of their absolute dimensions, and 2) the applied ripple predictors can predict the order of magnitude of ripple dimensions.

General comments

Overall the MS is well structured, well written and easy to read. KW outline that one

of the objectives is to report, and overall this contribution is more of a technical report. Nevertheless, I find it relevant and useful for the community. High-resolution and high-precision in situ measurements of small scale bedforms, whether ripples or dunes, are still relatively scarce, especially in deeper waters like shelf environments which are logistically challenging.

Moreover, the methodology is sound, although it should be better imbedded in already published methods in order to better distinguish between existing methods as opposed to newly developed methods.

The MS lacks explanations of the measured ripple dimensions in relation to the measured hydrodynamics and it also lacks explanations of the deviations in trends between measured and predicted ripple dimensions. Including this would inevitable improve the MS; however, this may require even further revision and may also be beyond the intended scope of the MS. I find the MS to fit within the scope of OS and I recommend pursuing publication; however, the MS would improve by a revision.

Below a list of suggestions of more overall and general character, which KW may consider for improving the MS:

1) Consider revising aim and objectives: See section comments for more details (comment to page 3, lines 4-9).

2) Consider elaborating further on morphological adaptation in non-steady conditions: See section comments for more details (comment to page 2, lines 32-34).

3) Consider elaborating further on system interference in near-bed lander setups: See section comments for more details (comment to page 3, lines 25-27).

4) Consider elaborating further on sampling strategies in non-steady conditions: See section comments for more details (comment to page 4, line 16).

5) Consider including more predictors in the comparison: See section comments for more details (page 7, lines 20-29).

Interactive
comment

6) Consider elaborating further on the observed and predicted ripple dynamics in relation to the observed hydrodynamics: See section comments for more details (comment to page 12, lines 20-22).

Comments and suggestions to each section of the MS are given below, including minor comments related to syntax and typos.

Section comments

Title

Consider including shelf environment in the title. Both because this is in essence a case study and because it qualifies the study to have determined ripple dimensions in a relatively deep water environment during both calm and wave conditions.

Abstract

The objectives listed in the abstract are not identical to the objectives outlined in the introduction. If listed in the abstract then they must be identical the objectives outlined in the introduction. In addition, the abstract must of course be updated in relation to a revised MS.

1 Introduction

Page 1, line 21: Consider changing "...as critical bed shear stress or dimensionless Shields parameter,..." to "...as the critical bed shear stress or the dimensionless Shields parameter,...".

Page 2, lines 3-5: Consider reformulating the sentence to something in the order of "In contrast to dunes, ripple dimensions are generally described as independent of the flow depth (see classification in Venditti, 2013); however, by applying a virtual boundary layer concept Bartholdy et al. (2015) recently demonstrated that water depth is actually a controlling factor along with grain size and flow velocity.".

Page 2, line 8: Consider also accrediting the seminal earlier works of Baas (1994)

demonstrating the time evolution of ripple dimensions in a flume study.

Page 2, line 16: Consider changing "..., assessment..." to "..., and assessment...".

Page 2, line 30: Consider changing "...unrelated..." to "...not related...".

Page 2, line 31: Consider changing "...bioturbation i.e., the..." to "...bioturbation, i.e. the...".

Page 2, lines 32-34: The time lag or duration of morphological adaptation in non-steady flow is a central issue. Consider elaborating on this by including earlier works on this topic as well as by including established and debated geomorphological concepts, e.g. process-materials-form, equilibrium, time-space scales, inheritance, and complexity and nonlinearity.

Page 3, line 1: Consider changing "...ripples i.e., bed..." to "...ripples, i.e. bed...".

Page 3, lines 4-9: The overall aim of the study is not specifically formulated, as opposed to the more specific objectives, which albeit are formulated slightly hidden within four sentences. From a taxonomy perspective the active verbs are describe, determine, derive, report, evaluate, compare and discuss. To some extent this outlines a stepwise development in the MS, which is clear and sound. Nevertheless, it could be improved in order to aid the reader. Consider outlining the overall aim of the study, and consider a more rigid and transparent formulation of the objectives. In order to raise the level of analysis consider adding an assessment in relation to the comparison (i.e. the comparison between measured and predicted dimensions); and also consider changing the discussion, which is a vague expression, to e.g. an evaluation or an assessment, or something in that line.

2 Methods

2.1 Study site

Page 3, line 12: Consider changing "...data was acquired..." to "...data were
acquired...". There is a standing debate on whether data are (or is) plural or singular; however, in general data are plural. I only mentioned it in this case, as I believe it is the first in the MS.

Page 3, line 12-19: The description of the study site settings is very limited. Consider presenting a location map that shows the location of the study site.

As KW applies a morphodynamic approach it would seem appropriate to outline and if possible visualize the environmental conditions of the system under investigation, e.g. the static or quasi-static boundary conditions like the overall geology, morphology (bathymetry) and sedimentology as well as the dynamic boundary conditions like the winds, waves and tides driving the hydro- and morphodynamics.

2.2 Lander deployments

Page 3, lines 25-27: Potential interference with the system under investigation is a central issue in any in situ measurements. Consider elaborating on this by including earlier works on this topic as well as by estimating and assessing a potential impact, e.g. in relation to the energy input to the system under investigation.

Page 3, line 28: What do environmental conditions refer to in this context? The term, and also environmental parameters (page 4, line 17), appears again later in the MS.

2.3 Devices and data

Page 4, line 16: In non-steady environment the duration of the individual measurement is a central issue. It is unclear from the text whether the 12 minutes interval of a full bathymetry scan, i.e. 5 scans per hour, also refers to a duration of 12 minutes per scan. Consider elaborating on the duration of each scan in relation to the dynamics of the seabed.

2.4 Bed detection methods

Page 5, lines 3-5: Seem to be a syntax issue. Consider rephrasing.

2.5 Coordinate conversion and gridding

Page 5, lines 23-33: This section is difficult to read due to the several symbols and numbers. In essence, however, it is simple trigonometry, so perhaps a schematic visualization could improve the readability. Hence, consider visualizing this section.

Page 5, line 33 to page 6, line 5: In general, approximations of higher elevations (like crests) can be determined quite good, whereas elevations of lower lying areas and small depressions (like troughs) are difficult to determine as the signal most likely gets reflected from the highest elevations within the ensonified area. However, here KW argue for the opposite. Please elaborate on this. As suggested above, it might also aid the reader to visualize this section.

Page 6, lines 6-7: Consider including the arguments for gridding the data at a cell size of 2.5 cm, i.e. arguing with the along and across track beam width as well as the overall point density.

2.6 Ripple geometry

Page 6, lines 19-20: In earlier works the relation between the stoss side length and the lee side length, to describe bedform asymmetry, has been termed symmetry index.

Page 6, lines 19-20: In earlier works the relation between the stoss side length and the lee side length, to describe bedform asymmetry, has been termed symmetry index.

Page 6, line 30 to page 7, line 18: The methods outlined for determining ripple dimensions display a mixture of continuous and discrete approaches. Similar methods and their advantages and disadvantages have been outlined and discussed in previous works by e.g. Robert (1988), Robert and Richards (1988), Nikora and Hicks (1997), Jerolmack and Mohrig (2005), Friedrich et al. (2007), Dijk et al. (2008), van der Mark and Blom (2007), van der Mark et al. (2008) (as also cited), Ernstsen et al. (2010). Consider elaborating on and discussing the apllied methods in relation to earlier works.

Page 7, line 8: Change "…(through)…" to "…(trough)…".

Page 7, line 8: Change "...transect..." to "...transects...".

2.7 Predictors for ripple dimensions

Page 7, lines 20-29: One of the key objectives (and part of the title) refers to a comparison between measured and predicted ripple dimensions. However, relatively few predictors are included in the analysis. It seems as if there are periods where only currents are mobilising the seabed. Hence, consider including additional predictors, so that all the different types of predictors considering input parameters are covered.

2.7.1 Current ripples

Page 8, line 16: Change "...are a valid..." to "...are valid...".

2.7.2 Wave ripples

2.8 Hydraulic roughness

Page 9, line 8: Change "...as is exceeds..." to "...as it exceeds...".

3. Results

3.1 Bed detection

OK.

3.2 Hydrodynamics

Page 10, line 1: Consider changing the subtitle to 3.2 Hydrodynamics and sediment mobility.

Page 10, lines 9-11: Referring to supercritical conditions in a section entitled hydrodynamics may easily be misunderstood as referring to supercritical flow conditions. Hence, consider instead to refer to e.g. excess shear stress or something in that line in order to improve readability and to avoid misunderstandings.

3.3 Ripple dimensions

Page 10, lines 13-32: Ripple lengths shown in Fig. 8b are not described in the results section; however, Fig. 8b is being referred to in the discussion. Nevertheless, consider also describing the measured and predicted ripple lengths in the results section with reference to Fig. 8b.

Page 10, line 20: Change "...returns..." to "...return...".

3.4 Hydraulic roughness

OK.

4 Discussion

4.1 Methods for dimension measurement

OK.

4.2 Precision of measurement

Page 11, lines 23-24: Seem to be a syntax issue. Consider rephrasing.

Page 11, line 24: Change "..., i.e., the..." to "..., i.e. the...".

Page 11, line 31 to page 12, line 2: One of the main advantages of a discrete approach for determining bedform dimensions is that it enables subsequent statistics on the distributions of bedform dimensions. Hence, consider showing these distributions e.g. as histograms along with the descriptive statistics. If showing the histograms then these should be included in the results section.

Page 12, line 4: Change "...predicted Soulsby et al. (2012)..." to "...predicted by Soulsby et al. (2012)...".

4.3 Form roughness

OK.

5 Conclusions

Page 12, lines 20-22: KW state that the observed dynamics of the ripple dimensions can be linked to changes in the forcing hydrodynamics. The time series are visualized in Fig. 7, however I don't recall any analysis and explanation of the variations. In addition, it seems as if the trend of the measured ripple height dynamics, after the peak in wave-related shear stress, is different from the trend of the predicted ripple height dynamics. Consider elaborating on this.

Page 12, line 21: Change "..., i.e., the relative changes can..." to "..., i.e. the relative changes, can...".

---

## Author Comment (AC1) · 1 Sep 2016

Response to Referee #1

Dear Referee,
thank you for the comments and suggestions for improvement of the manuscript. We have considered all recommendations and answer in detail below.

Please note that we found a slight inconsistency between the measured ranges shown in Fig. 7 and 8. this was corrected in the revised Fig. 8 (see last comment).

*Page 1*

*Line 9. I don't like "precision of detection". Either precision of some measurement or threshold of detection.*
Changed to "precision of measurement".

*Line 11-12. This sentence lacks mention of variations to which "up to a factor of two" applies.*
Context for "factor of two" added:
" ... by up to a factor of two between the traditional statistical estimate and a full evaluation of the spatial bathymetry."

*Line 20-21. "expressed . . parameter" gets in the way of the rest of this sentence. It could be a separate sentence (in parentheses).*
The sentence was rearranged with parentheses as suggested:
"If the threshold of motion (expressed as the critical bed shear stress or the dimensionless Shields parameter) for a characteristic grain size is exceeded, sediment is transported and bedforms develop."

*Page 2*
*Line 16. ". . modelling and assessment . ."*
Changed.

*Page 3*
*Somewhere near the end of the Introduction (top of page 3) should be a more explicit statement about the aim of this paper (before lines 4-9 saying what was done).*
The aims of the work have been stated more explicitly:
"The overall aims of this study are
1. An assessment of the precision of different methods for the detection and measurement of small scale bedforms from high resolution sonar data in a shelf sea environment
2. The comparison of the measurement precision to the dimensions of small scale bedforms calculated by different wave and current ripple predictors"

*Line 15. Unclear why Fig. 1 is referred to here.*
Unnecessary reference to Fig. 1 removed.

*Page 5*
*Line 3-5. This sentence is too complex and (I think) grammatically incorrect: "is exceeded" is redundant given the symbols "≥"? "respectively" is too far from the things being ordered. Break up the sentence, perhaps by defining $l_p$ and $l_{max}$ in a separate sentence.*
The sentence was rewritten as follows: "Threshold-level methods for bed detection in echo data acquired by similar sonars have been implemented by Smyth and Li (2005) and by Lefebvre (2009).

These authors detect the bed level at the depth where a certain percentage of the maximum ping-wise echo intensity $l_{max}$ is exceeded: $l_p \geq 0.6\, l_{max}$ (Smyth and Li, 2005) and $l_p \geq 0.8\, l_{max}$ (Lefebvre, 2009)."

*Line 6. Re-arrange (maybe split) this sentence. I think "where . . time" refers to the target shape and not to nadir.*
Part of the sentence was removed:
"These approaches are extended to account for the widening of the along-beam target shape with increasing grazing angles γ."

*Line 12,13. Echos do not have "slopes". Are you referring to the intensity as a function of distance as plotted in figure 3.*
Changed to: "... rising slope of the echo intensity signal."

*Line 20. "can be" – "was" if this is what you actually did. It reads like a good idea deserving careful description.*
Changed to show that the idea of a data-derived model bed echo was introduced here.

*Page 6*
*Line 4. I don't think η, λ have been defined; they could probably be replaced by words, or bring forward the definitions from line 16.*
Mentioning of dimension was removed here. Dimensions are introduced in "2.6 Ripple geometry".

*Line 15. The "phi" symbol should immediately follow "orientation".*
Changed.

*Line 30. "every deployment" is unclear. Probably not "deployment" but a briefest statement of what is the "ensemble".*
Changed to: "...for the complete deployment period".

*Page 7*
*Line 8. "(trough)" (typo).*
Corrected.

*Line 10-12.  This could be clearer. Is a crest the extreme maximum between any up-crossing and the next down-crossing of zero? If crests are defined dependent on zero-crossings, why not more directly use the average of distance between successive up-crossings of zero (or down-crossings of zero)? Presumably the result would be almost the same.*
The sentence was changes to clarify the procedure:
"The computed bedform height ηt is the average range between the elevations of detected maxima and minima per transect."
The use of average crest heights and trough depths makes removes the need to track successive crests and troughs within a transect.

*Line 28-29. So12w and So12c terminology implies separate predictors for ripples under waves and for ripples under currents. Then (line 29) "are used" but what exactly is applicable to mixed forcing conditions?*
The sentence was reformulated to clarify the procedure:

"For mixed forcing conditions, the recent wave and current ripple predictors of Soulsby et al. (2012) (So12w, So12c) are used by defining the prevailing dominant forcing and selecting the appropriate predictor."

*Page 8. Units need to be stated for the dimensionally inconsistent equations (4), (5).*
A sentence was added to clarify units for the empirical relation:
"SI-units are used in the equations for the following dimensionally inconsistent predictors."

*Page 11.*
*Line 4. ".. more pronounced than for Nikuradse roughness using (11); (12) results in .."*
Sentence rewritten for clarity:
"Due to the squared ripple height in Eq. 17, the difference between the methods is more pronounced for this than for Nikuradse's roughness using Eq. 16; and results in..."

*Line 23. This sentence is incorrect. Replace one "of" by ","?*
Corrected.

*Figure 8. There are two lines for Fl88 in (a) and Ya85 in (b). Please explain in caption – refer to (4), (5) and (3)?*
The figure was restructured. The two values for Fl88 correspond to mean and maximum height and the two values for Ya85 correspond to the range of 600-2000 $d_{50}$. This will be added to the figure caption.

---

## Author Comment (AC2) · 2 Sep 2016

Response to Referee #2

Dear Referee,
thank you for the detailed and in depth review and comments on this manuscript.
Following your suggestions, the methodology of bedform detection has been compared to previously published methods and advantages and disadvantages of the different methods applied have been highlighted.

Please note that we found a slight inconsistency between the measured ranges shown in Fig. 7 and 8. this was corrected in the revised Fig. 8 (see below).

*Below a list of suggestions of more overall and general character, which KW may consider for improving the MS:*
*1) Consider revising aim and objectives: See section comments for more details (comment to page 3, lines 4-9).*
*2) Consider elaborating further on morphological adaptation in non-steady conditions: See section comments for more details (comment to page 2, lines 32-34).*
*3) Consider elaborating further on system interference in near-bed lander setups: See section comments for more details (comment to page 3, lines 25-27).*
*4) Consider elaborating further on sampling strategies in non-steady conditions: See section comments for more details (comment to page 4, line 16).*
*5) Consider including more predictors in the comparison: See section comments for more details (page 7, lines 20-29).*
*6) Consider elaborating further on the observed and predicted ripple dynamics in relation to the observed hydrodynamics: See section comments for more details (comment to page 12, lines 20-22).*

1) Aim and objectives have been revised and stated more clearly, as also suggested by the first referee.
2) The concept of equilibrium and adaptation rates have been highlighted.
3) A passage about quality assessment of the data to exclude interference of the measuring platform has been added.
4) The sampling strategy has been described in relation to adaptation rates in non-steady conditions.
5) Two more wave and one more ripple predictors have been added to the comparison to complete the picture.
6) The comparison with predicted dimensions was chosen to be made for quasi stationary conditions, as most of the predictors provide equilibrium dimensions. The mismatch between the trend in predicted and measured ripple height is believed to be related to a migration event. However, the discussion of ripple dynamics is beyond the scope of this manuscript.

Detailed comments were addressed as follows.

*Title*
*Consider including shelf environment in the title. Both because this is in essence a case study and because it qualifies the study to have determined ripple dimensions in a relatively deep water environment during both calm and wave conditions.*
The title was changed to include the area of measurement more specifically:
"Predicted ripple dimensions in relation to the precision of in situ measurements in the Southern North Sea"

*Abstract*

*The objectives listed in the abstract are not identical to the objectives outlined in the introduction. If listed in the abstract then they must be identical the objectives outlined in the introduction. In addition, the abstract must of course be updated in relation to a revised MS.*

The abstract was updated to match the aims described in the introduction:

"Ripples are common morphological features in sandy marine environments. Their shapes and dimensions are closely related to local sediment properties and the forcing by waves and currents. Numerous predictors for the geometry and hydraulic roughness of ripples exist but due to their empirical nature, they may fail to properly reflect conditions in the field.

Here, situ measurements of ripple dimensions in a shallow shelf sea are reported. Discrete and continuous methods for the extraction of ripple dimensions from digital elevation models (DEM) are introduced. The range of measured ripple dimensions is quantified and compared to results of traditional and recent empirical predictors. The aims of this study are: 1. An assessment of the precision of different methods for the detection and measurement of small scale bedforms from high resolution sonar data and 2. a comparison of the measurement precision to the dimensions of small scale bedforms calculated by different wave and current ripple predictors. The precision of measurement of bedform dimensions is taken as the repeatability of a measurement for inactive conditions and the accuracy of measurement is assessed via comparison to predicted dimensions.

Results from field data show that the precision of measurement is limited to 10% of the absolute ripple dimensions and the order of magnitude of the ripple dimension can be predicted by the empirical relations. However, these tend to return the height of the largest ripples rather than average heights. The application of different methods for detection of heights may result in derived form roughness heights by up to a factor of two between the traditional statistical estimate and a full evaluation of the spatial bathymetry."

*1 Introduction*

*Page 1, line 21: Consider changing ". . .as critical bed shear stress or dimensionless Shields parameter,. . ." to ". . .as the critical bed shear stress or the dimensionless Shields parameter,. . .".*
Changed.

*Page 2, lines 3-5:  Consider reformulating the sentence to something in the order of "In contrast to dunes, ripple dimensions are generally described as independent of the flow depth (see classification in Venditti, 2013); however, by applying a virtual boundary layer concept Bartholdy et al. (2015) recently demonstrated that water depth is actually a controlling factor along with grain size and flow velocity.".*
Changed.

*Page 2, line 8: Consider also accrediting the seminal earlier works of Baas (1994) demonstrating the time evolution of ripple dimensions in a flume study.*
The reference for the works of Baas (1994) was added.

*Page 2, line 16:  Consider changing ". . ., assessment. . ." to ". . ., and assessment. . .".*
Changed.

*Page 2, line 30: Consider changing ". . .unrelated. . ." to ". . .not related. . .".*
Changed.

*Page 2, line 31: Consider changing ". . .bioturbation i.e., the. . ." to ". . .bioturbation, i.e. the. . .".*
Changed.

*Page 2, lines 32-34: The time lag or duration of morphological adaptation in non-steady flow is a central issue. Consider elaborating on this by including earlier works on this topic as well as by including established and debated geomorphological concepts, e.g. process-materials-form, equilibrium, time-space scales, inheritance, and complexity and nonlinearity.*

A section about adaptation equilibrium was added:

"The time required for the adaptation is a function of sediment transport rate and thus related to the excess shear stress induced by waves or currents and the grain size of the sediment. For current ripples, Baas (1994) showed in a flume study that the adaptation time is a function of the inverse power of flow velocity and ranges from a few minutes to several days. Additionally, bedform height is shown to adapt faster than wave length. His dataset was used to calibrate the empirical rate-of-change parameters in the time-evolving scheme by Soulsby et al. (2012) with two expressions for height and length. Nelson and Voulgaris (2014) stress that also wave-induced bedform height adapts last after wave length and orientation have almost reached a new stable equilibrium. The adaptation time for wave ripples is related to the wave period by Soulsby et al. (2012) and the rate-of-change parameter is related to the wave mobility number."

*Page 3, line 1: Consider changing ". . .ripples i.e., bed. . ." to ". . .ripples, i.e. bed. . .".*

Changed.

*Page 3, lines 4-9: The overall aim of the study is not specifically formulated, as opposed to the more specific objectives, which albeit are formulated slightly hidden within four sentences. From a taxonomy perspective the active verbs are describe, determine, derive, report, evaluate, compare and discuss. To some extent this outlines a stepwise development in the MS, which is clear and sound. Nevertheless, it could be improved in order to aid the reader. Consider outlining the overall aim of the study, and consider a more rigid and transparent formulation of the objectives. In order to raise the level of analysis consider adding an assessment in relation to the comparison (i.e. the comparison between measured and predicted dimensions); and also consider changing the discussion, which is a vague expression, to e.g. an evaluation or an assessment, or something in that line.*

As also suggested by the first referee, the aims of the study were stated more clearly:

"The overall aims of this study are

1. An assessment of the precision of different methods for the detection and measurement of small scale bedforms from high resolution sonar data in a shelf sea environment

2. The comparison of the measurement precision to the dimensions of small scale bedforms calculated by different wave and current ripple predictors"

The individual steps in the manuscript were described:

"In the following the bathymetry and sedimentary conditions at the study site on a sandy shelf seabed in the North Sea are described. The setup and devices used to measure the relevant data are shortly introduced. Processing steps for different methods to extract bedform dimensions from raw sonar data are detailed. The measured hydro- and morphodynamic data and ripple characteristics collected over two tidal cycles are analyzed.

The ranges and error margins determined by the technical specifications of the sensors and different methods employed to derive parameters from raw sensor data are reported. The range of bedform dimensions as a result of different methodology is shown and evaluated. This range is related and assessed with respect to the dimensions derived from ripple predictors. Implications for the calculation of bedform roughness from ripple dimensions are discussed."

*2 Methods*
*2.1 Study site*

*Page 3, line 12: Page 3, line 12: Consider changing ". . .data was acquired. . ." to ". . .data were acquired. . .". There is a standing debate on whether data are (or is) plural or singular; however, in general data are plural. I only mentioned it in this case, as I believe it is the first in the MS.*
Changed. "Data" is used in the plural form.

*Page 3, line 12-19: The description of the study site settings is very limited. Consider presenting a location map that shows the location of the study site. As KW applies a morphodynamic approach it would seem appropriate to outline and if possible visualize the environmental conditions of the system under investigation, e.g. the static or quasi-static boundary conditions like the overall geology, morphology (bathymetry) and sedimentology as well as the dynamic boundary conditions like the winds, waves and tides driving the hydro- and morphodynamics.*
An overview map of the German Bight with the location of the site was added to Figure 1.

[Figure]

The description of the study site has been updated:
"Station NOAH-D is located 40 km North of the East Frisian island Baltrum in a water depth of 35 m. Prior to deployment, a survey of the area surrounding the deployment site by multibeam echosounder revealed a flat and featureless bathymetry on the larger scale (500 m radius). The grain size analysis of grab samples taken prior to deployment of the lander showed bed sediments of fine sand with a median grain size d50 = 105 µm. Additional grab samples in the surrounding area exhibit spatially homogeneous sedimentary conditions which is supported by spatially homogeneous backscatter intensity in the multibeam data (not shown)."

A figure showing the grain size distribution was added.

[Figure]

Wind speed and direction were added to the hydrodynamic boundary conditions The meteorological data was obtained from the ship's weather station.

[Figure]

*2.2 Lander deployments*

*Page 3, lines 25-27: Potential interference with the system under investigation is a central issue in any in situ measurements. Consider elaborating on this by including earlier works on this topic as well as by estimating and assessing a potential impact, e.g. in relation to the energy input to the system under investigation.*

Potential interference with the observation platform was discussed.

"Minimization of interference with the system under investigation was a key factor in the design process of the lander as a benthic observatory. In contrast to tripod frames, the four-legged structure allows free flow between the legs. During the launch of the lander the heading is monitored to ensure orientation in alignment with the dominant bottom current direction with the help of a tail-fin on the launching frame. Bathymetry data are checked for the development of scour in vicinity of the legs and disregarded if the bathymetry in the central section is affected. However, with the current velocities common to the deployment sites in the open German Bight, such effects were not observed. Flow velocity and turbulence data are evaluated for possible influence by the lander frame or by other devices and removed if any influence is detected (Amirshahi et al., 2016)."

A photo of the lander during launch was added to show the launching frame with the tail-fin for positioning in the bottom current.

[Figure]

*Page 3, line 28: What do environmental conditions refer to in this context? The term, and also environmental parameters (page 4, line 17), appears again later in the MS.*
"Environmental conditions" were addressing properties of the seawater (salinity, temperature, turbidity). As these data are not evaluated, the sentence was removed.

*2.3 Devices and data*
*Page 4, line 16: In non-steady environment the duration of the individual measurement is a central issue. It is unclear from the text whether the 12 minutes interval of a full bathymetry scan, i.e. 5 scans per hour, also refers to a duration of 12 minutes per scan. Consider elaborating on the duration of each scan in relation to the dynamics of the seabed.*
The passage has been adapted to clarify the sampling interval and duration of the scans:
"... a full bathymetry scan was acquired in 11:50 minutes, therefore the scan interval, i.e. the sampling rate of the sonar, was set to 12 minutes."
A passage relating the scan interval and spatial resolution to observed dynamics of the seabed was added:
"Although not discussed here in detail, bedform migration with displacement rates of up to 3 cm per hour were observed. At a sampling rate of five scans per hour, this results
in a maximum migration distance of 0.6 cm between two successive scans which is lower than the selected resolution of the gridded small scale bathymetries."

*2.4 Bed detection methods*

*Page 5, lines 3-5: Seem to be a syntax issue. Consider rephrasing.*
As also suggested by the first referee, the sentence was reformulated:
"Threshold-level methods for bed detection in echo data acquired by similar sonars have been implemented by Smyth and Li (2005) and by Lefebvre (2009). These authors detect the bed level at the depth in which a certain percentage of the maximum ping-wise echo intensity $l_{max}$ is exceeded: $l_p \geq 0.6$ $l_{max}$ (Smyth and Li, 2005) and $l_p \geq 0.8$ $l_{max}$ (Lefebvre, 2009)."

*2.5 Coordinate conversion and gridding*
*Page 5, lines 23-33: This section is difficult to read due to the several symbols and numbers. In essence, however, it is simple trigonometry, so perhaps a schematic visualization could improve the readability. Hence, consider visualizing this section.*
A sketch was added to visualize the geometrical properties of the scanning sonar.

[Figure]

*Page 5, line 33 to page 6, line 5:  In general, approximations of higher elevations (like crests) can be determined quite good, whereas elevations of lower lying areas and small depressions (like troughs) are difficult to determine as the signal most likely gets reflected from the highest elevations within the ensonified area. However, here KW argue for the opposite. Please elaborate on this. As suggested above, it might also aid the reader to visualize this section.*

Correct. The section was adapted:

"As the acoustic pulse is most likely reflected by the highest elevation within the sonar footprint, the depth of troughs may be underestimated. Assuming a triangular bedform shape and a maximum slope equal to the angle of repose of sand $\alpha = 32\circ$, the maximum error in underestimating through depths yields $\varepsilon_{max} = 0.5 \cdot w_f \cdot \tan \alpha = 0.017$ m at nadir and $\varepsilon_{max} = 0.070$ m at the outermost beam in our configuration. As ripple troughs are usually more flat, the error is expected to be less pronounced. With a typical aspect ratio (ripple height over length) $\psi = 0.1$ much lower than the angle of repose, the maximum error reduces to $\varepsilon_{max} = 0.003$ m at nadir and $\varepsilon_{max} = 0.011$ m for the outermost ping."
The respective sketch was updated.

[Figure]

*Page 6, lines 6-7: Consider including the arguments for gridding the data at a cell size of 2.5 cm, i.e. arguing with the along and across track beam width as well as the overall point density.*
Arguments for grid spacing were included:

"For comparability among successive scans, the scattered data points were gridded resulting in digital elevation models (DEM) with consistent grid cells. With a minimum along-swath sonar step size of 0.028 m at nadir, a grid horizontal grid resolution of $\Delta x = \Delta y = 0.025$ m was selected to maintain the high resolution in the center of the recorded bathymetry evenif the effective resolution decreases with increasing beam footprint and spacing towards higher grazing angles."

*2.6 Ripple geometry*
*Page 6, lines 19-20: In earlier works the relation between the stoss side length and the lee side length, to describe bedform asymmetry, has been termed symmetry index.*
The term "symmetry index" was adopted.

*Page 6, line 30 to page 7, line 18: The methods outlined for determining ripple dimensions display a mixture of continuous and discrete approaches. Similar methods and their advantages and disadvantages have been outlined and discussed in previous works by e.g. Robert (1988), Robert and Richards (1988), Nikora and Hicks (1997), Jerolmack and Mohrig (2005), Friedrich et al. (2007), Dijk et al. (2008), van der Mark and Blom (2007), van der Mark et al. (2008) (as also cited), Ernstsen et al. (2010). Consider elaborating on and discussing the apllied methods in relation to earlier works.*
The methods were categorized into continuous and discrete/direct approaches. Advantages and disadvantages from the suggested literature are outlined:
"Methods for evaluation of bedform dimensions can be divided in continuous and discrete approaches. While statistical methods evaluate the continuum of the bathymetry, discrete or direct methods provide dimensions of a limited number of features detected with a given threshold for height and in case of the transect method also length. As described by Friedrich et al. (2007), the disadvantage of discrete methods is the sensitivity of measured dimensions to the thresholds selected. [...] Especially when primary and secondary bedforms are present, a carefully calibrated direct approach may be more useful than a statistical approach (cf. van der Mark et al., 2008; Ernstsen et al., 2010). The disadvantage of direct approaches is that the selection of thresholds and filter window sizes introduces a certain subjectivity and influences the resulting statistics of bedform dimensions. The advantage of the direct methods is that they capture a range of bedform dimensions and therefore yields not only average values for the overall bathymetry but also a distribution of dimensions which allows for a statistical evaluation."

*Page 7, line 8: Change ". . .(through). . ." to ". . .(trough). . .".*
Changed.

*Page 7, line 8: Change ". . .transect. . ." to ". . .transects. . .".*
Changed.

*2.7 Predictors for ripple dimensions*

*Page 7, lines 20-29:  One of the key objectives (and part of the title) refers to a comparison between measured and predicted ripple dimensions. However, relatively few predictors are included in the analysis. It seems as if there are periods where only currents are mobilising the seabed. Hence, consider including additional predictors, so that all the different types of predictors considering input parameters are covered.*
Two more wave-ripple predictors (Grant and Madsen, 1982; Li et al., 1996) and another current ripple predictor (Baas, 1994) were added. The corresponding figure was updated for the comparison between predicted and measured ripple dimensions.

*2.7.1 Current ripples*
*Page 8, line 16: Change ". . .are a valid. . ." to ". . .are valid. . .".*
Changed.

*2.7.2 Wave ripples*

*2.8 Hydraulic roughness*

*Page 9, line 8: Change ". . .as is exceeds. . ." to ". . .as it exceeds. . .".*
Changed.

*3 Results*
*3.1 Bed detection*
*OK.*

*3.2 Hydrodynamics*

*Page 10, line 1: Consider changing the subtitle to 3.2 Hydrodynamics and sediment mobility.*
Changed.

*Page 10, lines 9-11:  Referring to supercritical conditions in a section entitled hydrodynamics may easily be misunderstood as referring to supercritical flow conditions. Hence, consider instead to refer to e.g. excess shear stress or something in that line in order to improve readability and to avoid misunderstandings.*
The term "supercritical conditions" was substituted by a "excess shear stress":
"For the first 18 hours of the deployment, conditions with excess shear stress were observed only during peak flood and ebb current. Wave-induced excess shear stress conditions are reached for a period of 4 hours starting around 15:00 local time on the second day, followed by a period with current-induced excess shear stress lasting for around 4 hours during flood current."

*3.3 Ripple dimensions*
*Page 10, lines 13-32: Ripple lengths shown in Fig. 8b are not described in the results section; however, Fig. 8b is being referred to in the discussion. Nevertheless, consider also describing the measured and predicted ripple lengths in the results section with reference to Fig. 8b.*
Ripple lengths in Figure 8b are described.

*Page 10, line 20:  Change ". . .returns. . ." to ". . .return. . .".*
Changed.

*3.4 Hydraulic roughness*
*OK.*

*4 Discussion*
*4.1 Methods for dimension measurement*
*OK.*
*4.2 Precision of measurement*

*Page 11, lines 23-24: Seem to be a syntax issue. Consider rephrasing.*
The sentence was rephrased:
"To assess the accuracy of the measurement, a priori known topography under controlled laboratory conditions would be required."

*Page 11, line 24:  Change ". . ., i.e., the. . ." to ". . ., i.e. the. . .".*
Changed.

*Page 11, line 31 to page 12, line 2: One of the main advantages of a discrete approach for determining bedform dimensions is that it enables subsequent statistics on the distributions of bedform dimensions. Hence, consider showing these distributions e.g. as histograms along with the descriptive statistics. If showing the histograms then these should be included in the results section.*

A figure illustrating the evolution of the statistics of bedform dimensions from the transect method in combination with the descriptive statistics will be added to the revised manuscript.

[Figure]

*Time-stacked histograms of dimension measurements from the transect method. (a) Ripple height, (b) ripple length.*

*Page 12, line 4: Change ". . .predicted Soulsby et al. (2012). . ." to ". . .predicted by Soulsby et al. (2012). . .".*

Changed.

*Page 12, lines 20-22: KW state that the observed dynamics of the ripple dimensions can be linked to changes in the forcing hydrodynamics. The time series are visualized in Fig. 7, however I don't recall any analysis and explanation of the variations. In addition, it seems as if the trend of the measured ripple height dynamics, after the peak in wave-related shear stress, is different from the trend of the predicted ripple height dynamics. Consider elaborating on this.*

Ripple migration was observed during the second day of the deployment. The discussion of bedform migration and its effect on their dimensions is beyond the scope of this manuscript, as we compare measured dimensions in quasi-stationary conditions to predicted equilibrium dimensions. However, the fact that the trends of predicted and measured heights are opposed was added:

"Additionally, the migration of bedforms observed may result in the opposed trends in the development of ripple heights during the wave dominated conditions (around 18:00~h on the second day) (Fig. 7)."

*Page 12, line 21: Change ". . ., i.e., the relative changes can. . ." to ". . ., i.e. the relative changes, can. . .".*

Changed.